# Intellectual conflicts of interest among cardiology and pulmonology clinical practice guidelines

J. Henry Brems[1,2]*, Taylor Wagner[2], Julia Diamant[2], Andrea E. Davis[3], Ellen Wright Clayton[4,5]

1 Division of Pulmonary and Critical Care Medicine, Johns Hopkins University School of Medicine, Baltimore, MD, United States of America, 2 Berman Institute of Bioethics, Johns Hopkins University, Baltimore, MD, United States of America, 3 Department of Medicine, Vanderbilt University Medical Center, Nashville, TN, United States of America, 4 Center for Biomedical Ethics and Society, Vanderbilt University Medical Center, Nashville, TN, United States of America, 5 Law School, Vanderbilt University, Nashville, TN, United States of America

* jbrems2@jh.edu

## Abstract

### Background

Intellectual conflicts of interest (COI), like financial COI, may threaten the validity and trustworthiness of clinical practice guidelines (CPGs). However, comparatively little is known about intellectual COI in CPGs. This study sought to estimate the prevalence of intellectual COI and corresponding management strategies among cardiology and pulmonology CPGs.

### Methods

We conducted a retrospective document review of CPGs published by cardiology or pulmonology professional societies from the United States, Canada, or Europe from 2018 to 2019 available via the Emergency Care Research Institute, Guidelines International Network, or Medscape databases. We assessed the percentage of authors with an intellectual COI, defined as i) authorship on a study reviewed by the CPG, ii) authorship of a prior editorial related to a CPG recommendation, or iii) authorship of a prior related CPG. Management strategies assessed included use of GRADE methodology, inclusion of a methodologist, and recusals due to intellectual COI. Outcomes were assessed overall and compared between cardiology and pulmonology CPGs.

### Results

Among the 39 CPGs identified (14 cardiology, 25 pulmonology), there were a total of 737 authors, of whom 473 (64%) had at least one intellectual COI. Among all CPGs, a median of 67% (Interquartile Range 50%-76%) of authors had at least one intellectual COI, and COI was more prevalent among cardiology compared with pulmonology CPGs (84% vs 57%, p<0.001). There was variable use of management strategies among the CPGs, including use of GRADE methodology (64% of CPGs), inclusion of a methodologist (49%), and recusals due to intellectual COI (0%).

**Data Availability Statement:** All relevant data are within the paper and its Supporting Information files.

**Funding:** The author(s) received no specific funding for this work.

**Competing interests:** The authors have declared that no competing interests exist.

## Conclusion

Intellectual conflicts of interest appear to be highly prevalent and under-reported among cardiology and pulmonology CPGs, which may threaten their validity. Greater attention to and improved management of intellectual COI by CPG-producing organizations is needed.

## Introduction

Clinical practice guidelines (CPGs) provide critically important recommendations on the delivery of care throughout the medical field [1]. In its 2011 report, the Institute of Medicine (IOM), now known as the National Academy of Medicine, defined CPGs as producing recommendations on the basis of systematic reviews of the evidence [2]. Thus, compared with other forms of guidance statements which may rely more on expert opinion, CPGs place the highest emphasis on evidence-based recommendations.

Despite this emphasis on an evidence base, CPGs necessarily require some subjective interpretation by panel members and so are susceptible to conflicts of interest (COI). COI within CPGs have received significant attention in nearly every medical sub-specialty over the past two decades, but assessments have focused almost exclusively on financial COI [3–12]. However, non-financial COI, such as intellectual COI, also threaten the validity of guidelines [2].

While all individuals have pre-existing opinions, intellectual COI are defined as "academic activities that create the potential for an attachment to a specific point of view that could unduly affect an individual's judgment about a specific recommendation" [2, 13]. In the context of CPGs, a guideline panelist may have authored primary studies that are being reviewed by the guideline. That panelist is likely to believe the results of their own study and have an interest in promoting their own findings. Thus, they may be biased in their recommendation.

Intellectual COI may cause harm either by biasing recommendations or by producing guidelines that appear less trustworthy and thereby impair the uptake of well-founded recommendations. As an example of the latter, the American Academy of Family Medicine refused to endorse the 2017 American College of Cardiology (ACC)/American Heart Association (AHA) guideline on hypertension largely due to intellectual COI [14]. Conversely, other organizations have taken public steps to combat intellectual COI in CPG development. In particular, because content experts, while necessary, may be more likely to have intellectual COI, some organizations such as the American College of Chest Physicians (ACCP) have chosen to mitigate this issue by including more methodologists in the development process [15–17].

Despite growing concerns, the prevalence and management of intellectual COI among CPGs remains unclear [18]. A single study of two Japanese CPGs found that 27–47% of cited studies were authored by CPG panelists [18, 19]. However, it did not evaluate other suggested metrics of intellectual COI such as editorials or prior guidelines authorship, [13, 20] and it is unclear how representative those results are of intellectual COI in CPGs more broadly.

Thus, we sought to investigate the prevalence of intellectual COI and management strategies among CPGs. As a secondary objective, we also sought to investigate the variability in these measures among different organizations. Due to criticism received by the ACC/AHA as well as publicized measures taken by the ACCP, [14, 16] we investigated intellectual COI among recent cardiology and pulmonology CPGs.

## Materials and methods

### Sample selection

We included all CPGs published by cardiology or pulmonology professional societies in the United States, Canada, or Europe between January 1, 2018 and December 31, 2019. Given the clinical overlap between pulmonary and critical care medicine, we included critical care societies among the pulmonology group.

Because no comprehensive CPG database exists since the National Guidelines Clearinghouse became unavailable due to a loss of funding, [21] we searched three databases—Emergency Care Research Institute, Guidelines International Network, and Medscape—as described in a previous study [22]. For every potential guideline that met our initial inclusion criteria, we searched the corresponding professional societies' websites for any other guidelines. We excluded documents titled as 'focused updates,' 'position statements,' or 'consensus documents,' which were explicitly differentiated from CPGs by their organization. To further ensure all potential guidelines met the IOM definition of a CPG, they were included only if they contained: (i) a systematic review, (ii) an assessment of benefits and harms, and (iii) explicit recommendations [2]. Further, due to the lack of an existing comprehensive database, a two-year time frame was selected for this study for comparability to prior systematic evaluations of CPGs [18, 22].

As many CPGs are developed jointly by multiple professional societies, we classified each CPG as cardiology or pulmonology according to the society that was clearly distinguished as primary in the title, or if no clear designation existed, the society in whose associated journal the CPG was published. Other societies were considered as partner organizations, and data were only collected up to the first three organizations listed.

### Intellectual COI measures

We assessed intellectual COI among all CPG authors and only considered those explicitly listed as authors by the CPG. We did not evaluate others listed only as reviewers or panelists because these were non-standardized and their contribution to CPG development was not always clear.

For all CPG authors, we assessed whether they had any of the following intellectual COI:

i.  authorship on a study reviewed by the CPG

ii.  publication of a prior editorial

iii.  membership on a prior CPG panel

These criteria for intellectual COI have been used before, are quantifiable, and represent a potential strong prior attachment to a clinical viewpoint [13, 18, 19].

To determine authorship on a reviewed study, we cross-referenced each CPG author's PubMed bibliography with the CPG's references. To ensure any identified publications represented authorship of a study considered by CPG in formulating a distinct recommendation, we excluded CPG references that were cited only in the introduction or conclusion of the CPG. In addition, we required the identified publication be a clinical trial, cohort study, case-control study, case series, meta-analysis, and systematic review as these are the data sources generally considered by CPGs in formulating recommendations. Lastly, we ensured any identified studies had the same author as the CPG, defined by the same name plus institutional affiliation.

To determine publication of a prior editorial or membership on a prior CPG panel, two reviewers (JHB and TW or JD) conducted a search of each CPG author via PubMed. For

editorials, each reviewer independently searched the author's bibliography on PubMed using the 'Editorial' search criterion. An initial title and abstract review were used to exclude any documents that were irrelevant to the CPG or were not in fact editorials. Potentially relevant editorials were then reviewed to determine if they expressed a clear opinion for or against an explicit recommendation of the CPG. A similar process was used to identify prior panel membership, using the "Guideline" and "Practice Guideline" search criteria from PubMed. Prior editorials and CPGs did not have to be referenced by the current CPG to be included. Authors were identified by name and institutional affiliation as above. Any discrepancies were resolved via discussion among reviewers.

### Strategies for management of intellectual COI

We reviewed each CPG for multiple techniques to manage intellectual COI including use of the Grading and Recommendations, Assessment, Development, and Evaluation (GRADE) methodology, including methodologist(s) on the CPG panel, COI disclosures, and recusals due to COI. The GRADE methodology offers a systematic method for translating evidence into recommendations and has been used to manage intellectual COI [16, 23, 24]. Any COI disclosures and recusals were assessed directly from the guideline. We focused solely on non-financial disclosures and corresponding recusals.

We also assessed whether each organization had a policy in place for management of COI within CPGs and whether that policy specifically considered intellectual COI. This was conducted via a two-reviewer (JHB, AED) online search, as described previously [22].

### Composition of guideline panels

Lastly, we obtained information on the composition of CPG panels. Using each CPG author's name and affiliation, we conducted an internet search with www.Google.com to determine if they were: i) 'clinical expert'–defined as a sub-specialist physician in the same field as the CPG (i.e. a cardiologist on a cardiology CPG, ii) 'other physician'–defined as a physician not in the same field as the CPG, iii) a 'methodologist', iv) non-physician health care professional (HCP) (including RN and PhD), or v) non-HCP. With the exception of 'methodologist,' determination of each author's role was based on their title, degree, and departmental affiliation from their institutional webpage. Methodologists, who have specific expertise in guideline methodology that is not necessarily reflected by any degree, were considered those who were identified explicitly as such in the CPG itself.

We additionally searched every guideline to determine which, if any, authors were denoted as chairs, co-chairs, or vice-chairs. All were collectively categorized as 'chairs.'

### Analysis

Our primary outcome was the percentage of authors with any intellectual COI, defined as the percentage of authors among a CPG who were identified as having at least one of the three intellectual COI sub-types.

Descriptive statistics were generated for all sub-types of intellectual COI, strategies to manage intellectual COI, and the composition of CPG panels. Measures of intellectual COI were summarized by chairs and by all authors (inclusive of chairs/co-chairs). Additionally, frequency of intellectual COI was summarized by author role. Summary statistics were generated among all CPGs as well as separately among cardiology and pulmonology CPGs.

To analyze the variation in intellectual COI by specialty, we compared the percentage of authors with any intellectual COI among cardiology and pulmonology guidelines using Mann

Whitney U test. We repeated this analysis for all sub-types of intellectual COI and additionally compared management strategies by organization type using Chi-squared test.

Kappa statistic for inter-rater reliability was calculated for the two-reviewer search used to identify editorials and CPGs. For the two-reviewer search of prior editorial or CPG panel membership, our reviewers demonstrated 85.3% agreement. Cohen's kappa statistic was 0.65 (95% Confidence Interval 0.60–0.70).

All data analyses were conducted using Microsoft Excel and Stata v17.0.

## Results

### Guideline and organization characteristics

We identified a total of 39 CPGs produced by 16 cardiology or pulmonology professional societies published between January 1, 2018 and December 31, 2019 (S1 Fig). Of the CPGs, 14 (31%) were produced by cardiology organizations and 25 (69%) by pulmonology organizations. A single CPG was produced by a cardiology organization with a pulmonology organization as a partner. Of note, 35 (90%) of the CPGs were primarily produced by 6 organizations. The numbers of CPGs produced by each organization are shown in Table 1, and all CPGs are listed in S1 Table.

### Prevalence of intellectual conflict of interest

Overall, there were a total of 737 authors, of whom 473 (64.2%) of authors had at least one intellectual COI. All 39 (100%) CPGs had at least one author with an intellectual COI, and only 9 (23%) CPGs had fewer than 50% of authors with an intellectual COI. The most frequent type of intellectual COI was authorship on a reviewed study with a median of 44% (IQR 33–71%) of authors having such a COI. Full results by intellectual COI sub-type are shown in Table 2.

**Table 1. Organizations and number of CPGs produced.**

| | No. of Clinical Practice Guidelines (N = 39) | | |
|---|---|---|---|
| | **Primary Org.** | **Partner Org.** | **Overall** |
| **Pulmonary** | | | |
| American Thoracic Society | 9 | 0 | 9 |
| American College of Chest Physicians | 8 | 0 | 8 |
| British Thoracic Society | 3 | 1 | 4 |
| Cystic Fibrosis Foundation | 1 | 0 | 1 |
| European Respiratory Society | 1 | 3 | 4 |
| European Society of Intensive Care Medicine | 1 | 0 | 1 |
| Japanese Respiratory Society | 0 | 1 | 1 |
| Society of Critical Care Medicine | 1 | 0 | 1 |
| Society of Thoracic Surgeons | 0 | 1 | 1 |
| **Cardiology** | | | |
| American Heart Association | 3 | 2 | 5 |
| American College of Cardiology | 2 | 2 | 4 |
| European Association for Cardio-Thoracic Surgery | 0 | 1 | 1 |
| European Atherosclerosis Society | 0 | 1 | 1 |
| European Society of Cardiology | 9 | 0 | 9 |
| European Society of Hypertension | 0 | 1 | 1 |
| Heart Rhythm Society | 0 | 2 | 2 |

**Table 2. Frequencies of authors' intellectual COI by subtype, overall and by specialty.**

|  | All | Cardiology | Pulmonology | P-value* |
|---|---|---|---|---|
| Authorship (any) | 44 [33–71] | 74 [42–76] | 43 [29–64] | 0.004 |
| Authorship (first or last) | 29 [20–48] | 39 [26–61] | 25 [17–43] | 0.02 |
| Editorial | 23 [5–34] | 35 [22–44] | 17 [0–25] | 0.001 |
| Agrees with CPG | 20 [5–29] | 28 [20–44] | 17 [0–2] | 0.001 |
| Disagrees with CPG | 0 [0–4] | 2 [0–6] | 0 [0–1] | 0.17 |
| CPG | 33 [20–53] | 52 [43–64] | 24 [13–37] | <0.001 |
| Any iCOI | 67 [50–76] | 84 [68–88] | 57 [41–69] | <0.001 |

Frequencies are summarized as median [interquartile range]

*P-values calculated from Mann-Whitney U test of identical frequencies in cardiology and pulmonology CPGs

A total of 36 CPGs noted which author was the chair or co-chair, and 34 of 36 (94%) had at least one chair or co-chair with an intellectual COI. Among all CPGs, a median of 100% (IQR 100–100%) of chairs and co-chairs had an intellectual COI. Authorship on a reviewed study was also most common among chairs and co-chairs, with a median of 100% (IQR 50–100%) of chairs. The total number of authors and chairs with an intellectual COI identified by CPG is shown in S1 Table.

Intellectual COI was more frequent among cardiology compared with pulmonology CPGs (Fig 1). A median of 84% (IQR 68–88%) of authors on cardiology CPGs had an intellectual COI compared with a 56% (41–69%) of authors on pulmonology CPGs (p< 0.001). Similarly, all sub-types of intellectual COI were more common among cardiology CPGs, including authorship on reviewed studies (74 vs 43%, p = 0.004), first or last authorship of reviewed studies (39 vs 25%, p = 0.02), publication of a prior editorial (35 vs 17%, p = 0.001), and membership on prior CPG (52 vs 24%, p<0.001) (Table 2).

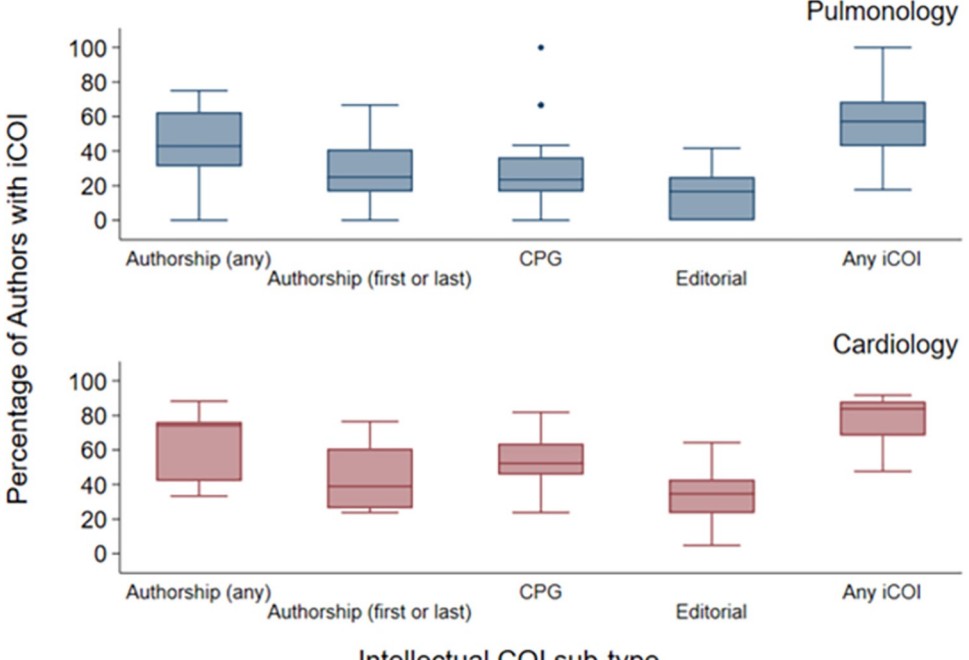

**Fig 1. Frequencies of intellectual COI by specialty.** Boxplot representing the percentage of authors identified with each sub-type of intellectual conflict of interest (iCOI) among (A) pulmonology and (B) cardiology CPGs.

**Table 3. Strategies for management of intellectual COI, overall and by specialty.**

|  | All (N = 39) | Cardiology (N = 14) | Pulmonology (N = 25) | P-value* |
|---|---|---|---|---|
| Use of GRADE methodology | 25 (64%) | 0 (0%) | 25 (100%) | <0.001 |
| Inclusion of methodologist | 19 (49%) | 0 (0%) | 19 (76%) | <0.001 |
| Any non-financial COI disclosed by authors | 17 (44%) | 11 (79%) | 6 (24%) | 0.003 |
| Funding source listed | 16 (41%) | 1 (7%) | 15 (60%) | 0.003 |
| Author(s) recused for any reason | 4 (10%) | 2 (14%) | 2 (8%) | 0.47 |
| Author(s) recused due to intellectual COI | 0 (0%) | 0 (0%) | 0 (0%) | 1.0 |
| Organizational policy addressing intellectual COI in CPGs | 18 (46%) | 0 (0%) | 18 (72%) | <0.001 |

*P-values calculated using Chi-squared test

## Management of intellectual conflict of interest

The frequency of each strategy to manage intellectual COI is shown in Table 3. While most strategies were seen in 40% or more of CPGs, only 4 (10%) CPGs had any author recusals, and no recusals were due to intellectual COI.

The use of strategies varied by specialty, as use of GRADE methodology (0 vs 100%, p<0.001) and inclusion of methodologists (0 vs 76%, p <0.001) were less common among cardiology compared with pulmonology CPGS. Moreover, 79% of cardiology CPGs had at least one author disclose a non-financial COI compared with 24% of pulmonology CPGs (p = 0.003).

## Composition of guideline panels

Among all CPGs, there was a median of 19 (IQR 14–22) panelists on the CPG. Clinical experts generally made up a majority of CPG panels with a median (IQR) of 71% (53–88%), and it was the only role present in every CPG. A total of 32 (82%) CPGs included other physicians, and 19 (49%) included a methodologist. Non-physician HCPs were included on 23 (59%) CPG panels, and non-HCPs were included in 13 (33%). The frequencies of professions on CPG panels are shown in Fig 2.

Overall, intellectual COI was most common among clinical experts, with 71% of such authors having at least one intellectual COI. Intellectual COI was least prevalent among methodologists and non-HCPs, with 14% of each having at least one COI. Frequencies of intellectual COI by author role are shown in Table 4.

## Discussion

In this retrospective study of cardiology and pulmonology CPGs, we found a high prevalence of intellectual COI with variable use of strategies to manage it. Our findings raise several important points.

First, the prevalence of intellectual COI is high and is comparable to financial COI. We found that 64% of all authors had an intellectual COI, which is similar to studies of financial COI, which occur among 40–80% of CPG authors [3–11, 18]. Even without accounting for financial COI, the frequency of intellectual COI alone means that more than three-quarters of CPGs investigated in this study fell short of the IOM standards for trustworthy CPGs, which state that a minority of CPG authors should have any COI [2].

While the potential for intellectual COI to bias CPG recommendations is unknown, our findings are nonetheless cause for substantial concern. Notably, the most common intellectual COI was authorship on a reviewed study, which has been previously recognized as an

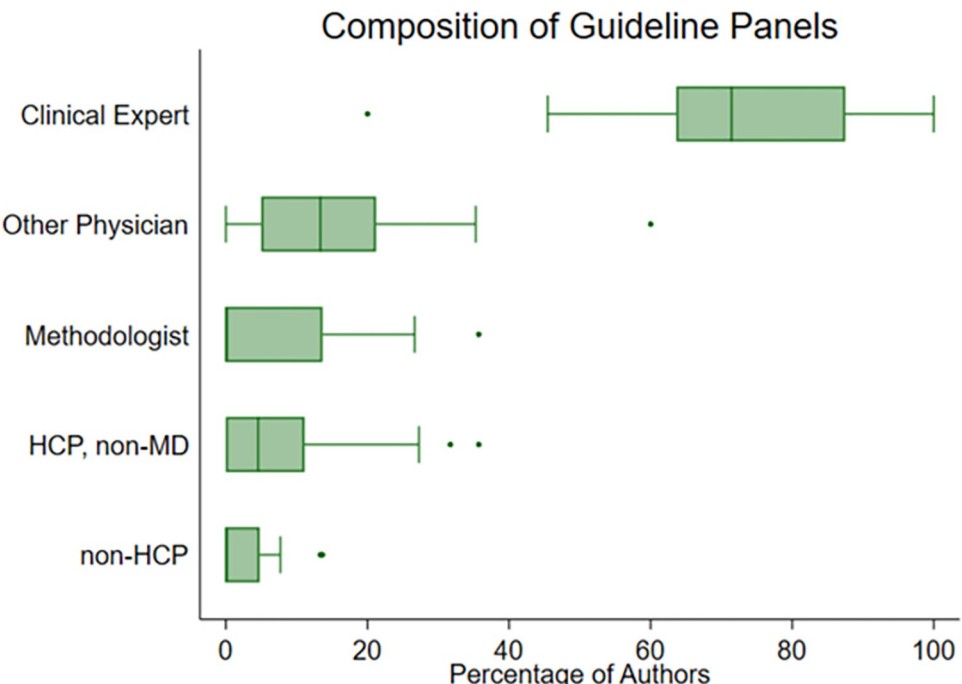

**Fig 2. Composition of guideline panels.** Boxplot of the percentage of authors with each role among all CPGs.

"important intellectual COI" [13]. Researchers have an interest in promoting their own findings, and intuitively, authors of original research should be expected to more strongly believe their results compared to a neutral observer. Within medical research more broadly, there are clear suggestions that intellectual biases impact clinicians' interpretation of evidence, as demonstrated by beliefs that persist despite contradicted findings, [25] the impact of favorable recommendations from like-minded peer reviewers, [26–28] and researchers' allegiance to their findings [29]. Although there is a lack of evidence, it has been suggested before that CPG panels including a member with intellectual COI could have bias in 15% of their recommendations [30]. While that estimate could be incorrect, even 5% of recommendations' being inappropriately biased could result in significant harm to patients given the high number of CPGs combined with the prevalence of intellectual COI.

Important differences between intellectual COI and financial COI should be considered in interpreting our results. First, financial COI likely only biases individuals in favor of the intervention for which they have a conflict. However, intellectual COI may bias an individual either for or against an intervention depending on their specific conflict (e.g. whether their prior

**Table 4. Percentages of authors with intellectual COI, by author role.**

| | Frequency of Intellectual COI, % | | | | |
|---|---|---|---|---|---|
| | Authorship (any) | Authorship (first or last) | Editorial | CPG | Any |
| **Author Role** | | | | | |
| Clinical Expert (N = 523) | 58% | 39% | 29% | 41% | 71% |
| Other Physician (N = 85) | 43% | 31% | 18% | 16% | 47% |
| Methodologist (N = 42) | 6% | 5% | 7% | 10% | 14% |
| Non-physician HCP (N 66) | 39% | 24% | 15% | 26% | 52% |
| Non-HCP (N = 21) | 10% | 10% | 0% | 5% | 14% |

study found evidence supporting or refuting an intervention). Second, in practice, financial COI disclosures are often only requested from the past few years, although the exact time frame varies by organization [31]. We did not impose a time limit for intellectual COI in this study because it is unclear if or when intellectual COI truly become irrelevant. Particularly because CPGs may consider evidence from more than a decade prior, intellectual COI have potential to endure. Despite these differences between financial and intellectual COI, both represent a threat to the validity of CPGs, are recognized by the IOM, and merit explicit consideration from CPG developers [2].

Regardless of the extent to which intellectual COI bias recommendations, they make CPGs less trustworthy. Intellectual COI are recognized as undermining trust by the IOM, [2] and the major family medicine professional society in the U.S. cited intellectual COI as grounds not to endorse an update in the ACC/AHA hypertension guidelines that would have doubled the number of patients under 45 diagnosed with hypertension [14, 32]. Assuming the ACC/AHA recommendations were optimally evidence-based, the presence of intellectual COI (even if it did not result in bias) still may have harmed patients by impairing uptake of the guidelines among family medicine practitioners. Trustworthiness of guidelines is of paramount importance.

Our results also address previous concerns surrounding inclusion of intellectual COI in CPG management. Some have previously argued that "non-financial COI" is too broad and is used to represent intellectual beliefs rather than to define actual conflicts of interests [33–35]. While these concerns remain valid for the term "non-financial COI," our results demonstrate that the narrower "intellectual COI" can be well-specified with only a few criteria and is highly prevalent.

Importantly, intellectual COI, as defined in our study, plausibly present a true COI rather than just intellectual beliefs because CPG panelists may have an interest in promoting their prior published works via CPG recommendation given the potential career and reputational benefits. The criteria used in our study cannot identify all potential intellectual COI or pre-existing viewpoints. However, they present a simple, concrete method that CPG-developers could use to better evaluate and manage intellectual COI. Attempts to identify and manage intellectual COI do not (and should not) need to identify all individuals with any pre-existing belief or opinion; identifying the highest-risk and verifiable conflicts would be a major step forward.

Further, CPG-producing organizations can and should take steps to better manage intellectual COI. The variability in management strategies between cardiology and pulmonology CPGs can be viewed as evidence that each of the strategies is practical yet needs to be more widely implemented. Crucially, recusals for intellectual COI ought to be strongly considered for a panelist reviewing their own work, but they were absent in our study. While improving disclosures of intellectual COI is also needed, disclosure alone is not sufficient and can worsen bias if not paired with appropriate management [36].

Our finding that clinical experts were the most likely participant to have an intellectual COI suggests that CPG developers must balance clinical expertise to mitigate intellectual COI. Since clinical expertise is necessary to interpret data and to make a clinical assessment of risks and benefits, it should not be eliminated for the sake of eliminating all COI. Rather, CPG developers should increase the diversity of panel members, limiting the number of panelists who have an intellectual (or any other relevant) COI to a minority of the panel. All panel members should be excluded from sections in of the CPG in which they would be reviewing their own studies.

In addition, because CPGs rely on a systematic review of the evidence, they ought to give more weight to methodologic expertise by increasing inclusion of methodologists. Intellectual

COI was far less common among methodologists in our study, and prior work suggests their inclusion impacts strength of recommendations [16]. Finally, while some important questions certainly require an even greater degree of clinical expertise due to a lack of evidence, those are best answered by other forms of guidance documents, which may explicitly rely on expert opinion rather than in a CPG.

It should be noted that, although we considered CPGs published by Canadian professional societies eligible for inclusion our study, we did not identify any that met full inclusion criteria. Given the lack of an existing single, universal CPG database, this could reflect a limit of our sampling strategy, which did not include a database such as the Canadian Medical Association's Infobase that may be more enriched for Canadian CPGs [37]. There may be important differences between Canadian, American, and European professional societies with respect to intellectual COI. Further studies are needed to confidently draw any conclusions about intellectual COI in Canadian CPGs.

Additionally, while we evaluated CPGs published in 2018 and 2019, some recent publications indicate that CPG developers may be paying increased attention to intellectual COI [38, 39]. The ACC—a major developer of cardiology CPGs—initiated a review of its CPG policies and procedures in 2020 [40]. While intellectual COI was not mentioned explicitly in that report, [40] it is possible that the COI policy or composition of CPG panels could be shifting for the ACC and for CPG developers more broadly. Thus, future studies to evaluate shifts in prevalence and management of intellectual COI over time will be critical.

Our study has multiple other limitations. First, we only assessed cardiology and pulmonology CPGs, which may not be representative of broader CPGs. Second, we did not investigate financial COI, so we cannot directly assess the prevalence or importance of financial COI relative to intellectual COI. However, even for financial COI, the thresholds used to define 'significant' or 'allowable' COI are arbitrary [41]. Given how common intellectual COI was in our study, CPG developers ought to consider and implement thresholds for it as well, even if further study is needed to refine those thresholds. Third, given the partly subjective nature of assessing prior editorials and CPGs, our estimates are likely imperfect. However, our interrater reliability demonstrated moderate agreement, and the majority of intellectual COI found in our study was based on authorship of reviewed studies, which is a more objective measure.

In conclusion, intellectual conflicts of interest appear to be highly prevalent and underreported among cardiology and pulmonology CPGs, which may threaten their validity. Further research is needed to understand the impact of intellectual COI on guideline recommendations, but regardless, greater attention to this issue, increasing recusal, and expanding the diversity of these panels, particularly in the area of methodology, can be implemented now.

## Supporting information

**S1 Table. Guidelines, professional societies, and intellectual conflicts of interests.**
(PDF)

**S1 Fig. Identification and assessment of clinical practice guidelines (CPGs).**
(TIF)

## Author Contributions

**Conceptualization:** J. Henry Brems, Ellen Wright Clayton.

**Data curation:** J. Henry Brems, Taylor Wagner, Julia Diamant, Andrea E. Davis.

**Formal analysis:** J. Henry Brems, Taylor Wagner, Julia Diamant, Andrea E. Davis.

**Investigation:** J. Henry Brems, Ellen Wright Clayton.

**Methodology:** J. Henry Brems, Ellen Wright Clayton.

**Supervision:** Ellen Wright Clayton.

**Writing – original draft:** J. Henry Brems.

**Writing – review & editing:** J. Henry Brems, Taylor Wagner, Julia Diamant, Andrea E. Davis, Ellen Wright Clayton.

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
