## [Decision Letter · Decision Letter 0]

15 Feb 2023

PONE-D-22-33083Intellectual conflicts of interest among cardiology and pulmonology clinical practice guidelinesPLOS ONE

Dear Dr. Brems,

Thank you for submitting your manuscript to PLOS ONE. After careful consideration, we feel that it has merit but does not fully meet PLOS ONE’s publication criteria as it currently stands. Therefore, we invite you to submit a revised version of the manuscript that addresses the points raised during the review process.

We look forward to receiving your revised manuscript.

Kind regards,

Gladys Honein-AbouHaidar

Academic Editor

PLOS ONE

Journal Requirements:

Additional Editor Comments:

Dear Dr. Brems,

We apologize for the delay in providing a feedback. We faced several rejections from reviewers. Hence, to avoid further delays, I acted as the second reviewer as well as the Academic Editor. Herein the second reviewer/ academic editor's comments on the study.

It is an important study focusing on the prevalence of intellectual conflict of interest among cardiology and pulmonology clinical practice guidelines. The topic is of paramount importance and has several consequences on patient outcomes.

The method is well described and solid.

Herein my suggestions to propose the quality of the study.

The study covered CPGs published up to December 2019. Since then, different organizations emphasized the importance of addressing non-financial conflict of interest in CPGs to cite a couple: Alhazzani W, Lewis K, Jaeschke R, Rochwerg B, Møller MH, Evans L, Wilson KC, Patel S, Coopersmith CM, Cecconi M, Guyatt G, Akl EA. Conflicts of interest disclosure forms and management in critical care clinical practice guidelines. Intensive Care Med. 2018 Oct;44(10):1691-1698. & Nejstgaard CH, Bero L, Hróbjartsson A, Jørgensen AW, Jørgensen KJ, Le M, Lundh A. Conflicts of interest in clinical guidelines, advisory committee reports, opinion pieces, and narrative reviews: associations with recommendations. Cochrane Database Syst Rev. 2020 Dec 8;12(12. One would expect that disclosing intellectual COI would better addressed by methodologists and panelists of CPGs. It will be of an added value to examine whether recent CPGs up to 2022 had better management of COIs.

The authors indicated USA, Canada and Europe will be included in the CPG search. From the findings, none of the CPGs was Canadian. Were there no CPGs during that period or they were not included in the search?

Additional Editor Comments:

In your Data Availability statement, you have specified that the data is fully available without restrictions. PLOS journals require authors to make all data necessary to replicate their study’s findings publicly available without restriction at the time of publication. When specific legal or ethical restrictions prohibit public sharing of a data set, authors must indicate how others may obtain access to the data. For more information about our data policy, please see http://journals.plos.org/plosone/s/data-availability.

Upon re-submitting your revised manuscript, please upload your data set including authors of CPGs as either Supporting Information files or to a stable, public repository and include the relevant URLs, DOIs, or accession numbers within your revised cover letter. For a list of acceptable repositories, please see http://journals.plos.org/plosone/s/data-availability#loc-recommended-repositories. Any potentially identifying patient information must be fully anonymized.

Reviewers' comments:

Reviewer's Responses to Questions

**Comments to the Author**

1. Is the manuscript technically sound, and do the data support the conclusions?

Reviewer #1: Yes

2. Has the statistical analysis been performed appropriately and rigorously? 

Reviewer #1: N/A

3. Have the authors made all data underlying the findings in their manuscript fully available?

Reviewer #1: No

4. Is the manuscript presented in an intelligible fashion and written in standard English?

Reviewer #1: Yes

5. Review Comments to the Author

Reviewer #1: This manuscript looks at intellectual COI in cardiology and pulmonology guidelines produced by societies in Canada, Europe and the United States between 2018-2019.

Although the authors mention studies by Bero and Grundy (their references 32 and 33) in relation to the ability to quantify intellectual COI, they do not deal with the core of Bero’s and Grundy’s arguments that nonfinancial influences/interests may not fit a definition of conflicts of interest. Further, Bero and Grundy propose other methods, such as reflexivity as a tool that can be used that makes transparent and accounts for researchers' professional and personal identities. The authors need to deal in more detail with whether there is a distinction between COI and interests/influences beyond just saying that both financial and intellectual COI can be quantified.

The authors declare that they don’t have any conflicts of interest to disclose but does that include intellectual COIs? For example, three of the authors (Helms, Davis and Clayton) published an article in PLoS One last year on COI policies among organizations producing CPGs. Does that constitute an intellectual COI?

Why did the authors choose the January 1, 2018 to December 31, 2019 time frame?

Since the authors were potentially including Canadian CPGs, why didn't they search the database of CPGs maintained by the Canadian Medical Association - https://joulecma.ca/cpg/homepage?_gl=1*u8a149*_ga*NTU2NTk4MzgzNzI2MTA5MDQuMTI1MTM5MzczNg..*_ga_91NZ7HZZ51*MTY3MTY0NzY4My41LjEuMTY3MTY0NzY4NS41OC4wLjA?

How was authorship defined? For example, if there were CPG committee members named within the guideline but not explicitly listed as authors were they considered as authors? What about internal reviewers of the guidelines, were they considered as authors?

On page 12, the authors mention the percent of authors with one of 4 intellectual COI subtypes but earlier they only give three subtypes: i) authorship on a study reviewed by the CPG; ii) publication of a prior editorial; iii) membership on a prior CPG panel.

Did the authors investigate whether CPG authors also had financial conflicts of interest? If both intellectual and financial COIs existed which one would have more of an effect or is that something that the CPG readers would have to determine?

On page 16 (4th line) the authors say that one paper suggested that 15% of the recommendations of a CPG could have been biased by intellectual COI but Akl et al do not provide any information about how that figure was derived.

6. PLOS authors have the option to publish the peer review history of their article (what does this mean?). If published, this will include your full peer review and any attached files.

Reviewer #1: **Yes: **Joel Lexchin

---

## [Author Response · Author response to Decision Letter 0]

22 Mar 2023

Thank your for the helpful comments and opportunity to revise our manuscript based off of them. We feel this has substantially improved the quality of our manuscript, and we have provided a point-by-point response in the attached 'Response to Reviewers' letter.

---

## [Decision Letter · Decision Letter 1]

12 Jun 2023

PONE-D-22-33083R1Intellectual conflicts of interest among cardiology and pulmonology clinical practice guidelinesPLOS ONE

Dear Dr. Brems,

Thank you for submitting your manuscript to PLOS ONE. After careful consideration, we feel that it has merit but does not fully meet PLOS ONE’s publication criteria as it currently stands. Therefore, we invite you to submit a revised version of the manuscript that addresses the points raised during the review process.

ACADEMIC EDITOR:

 It is highly required to address the reviewer's comments as they are meant to shed the light on important nuances between financial and intellectual COI. 

As for whether authors should declare their past publication on COI in CPGs as a COI, it is recommended but not required. 

We look forward to receiving your revised manuscript.

Kind regards,

Gladys Honein-AbouHaidar

Academic Editor

PLOS ONE

Reviewers' comments:

Reviewer's Responses to Questions

**Comments to the Author**

1. If the authors have adequately addressed your comments raised in a previous round of review and you feel that this manuscript is now acceptable for publication, you may indicate that here to bypass the “Comments to the Author” section, enter your conflict of interest statement in the “Confidential to Editor” section, and submit your "Accept" recommendation.

Reviewer #1: (No Response)

2. Is the manuscript technically sound, and do the data support the conclusions?

Reviewer #1: Partly

3. Has the statistical analysis been performed appropriately and rigorously? 

Reviewer #1: Yes

4. Have the authors made all data underlying the findings in their manuscript fully available?

Reviewer #1: Yes

5. Is the manuscript presented in an intelligible fashion and written in standard English?

Reviewer #1: Yes

6. Review Comments to the Author

Reviewer #1: The changes made by the authors are appreciated but I still have some remaining concerns.

The authors appear to be equating financial COI and intellectual COI creating a false equivalence. While both types of COI can lead to biases in CPGs, biases due to financial COI are all in the same direction, i.e., in favour of the product(s) being considered, whereas the same is not true of intellectual COI. This type of COI can bias decision either in favour or against the product(s) being considered.

Despite equating the two types of COI, the authors may be treating intellectual COI differently from financial COI. When financial COI is investigated it only typically covers the previous three years on the assumption (possibly incorrect) that interactions longer ago than 3 years will no longer bias people. In this study, in the case of intellectual COI, the authors have not put any time limit on when individuals put forward a point of view that could constitute an intellectual COI. Therefore, we do not know whether the COI occurred last year or 10 years ago.

The authors need to acknowledge that the absence of one of the three types of intellectual COI does not mean that other CPG authors did not have specific points of view about the issues being considered in the CPGs. Financial COIs are much easier to discover than intellectual COIs, since the latter will only be revealed if the individual has published or presented a particular point of view.

The authors should explicitly acknowledge that they did not search the Canadian Medical Association’s database of 1700 CPGs. They offer two explanations for this absence. First, they say that they were not aware of the database from their review of prior studies on COI within CGPs. However, the CMA database is mentioned in at least two relatively recent peer reviewed studies (Elder et al. CMAJ 2020;192:e617-e625; Shnier et al. BMC Health Services Research 2016;16:383). Second, claiming that any relevant Canadian CPGs would have been included in the three databases that they did search is speculation.

I will leave it to the editors to decide if the authors should declare their past publication on COI in CPGs as a COI.

7. PLOS authors have the option to publish the peer review history of their article (what does this mean?). If published, this will include your full peer review and any attached files.

Reviewer #1: **Yes: **Joel Lexchin

---

## [Author Response · Author response to Decision Letter 1]

13 Jun 2023

Thanks for you the opportunity to revise our submission. We have incorporated the feedback, and most notably, have added to our discussion to highlight the nuances between financial and intellectual COI. Please see our 'Response to Reviewers' letter for point-by-point feedback.

---

## [Decision Letter · Decision Letter 2]

26 Jun 2023

Intellectual conflicts of interest among cardiology and pulmonology clinical practice guidelines

PONE-D-22-33083R2

Dear Dr. Brems,

We’re pleased to inform you that your manuscript has been judged scientifically suitable for publication and will be formally accepted for publication once it meets all outstanding technical requirements.

Kind regards,

Sascha Köpke

Academic Editor

PLOS ONE

---

## [Editor Report · Acceptance letter]

30 Jun 2023

PONE-D-22-33083R2 

Intellectual conflicts of interest among cardiology and pulmonology clinical practice guidelines 

Dear Dr. Brems:

I'm pleased to inform you that your manuscript has been deemed suitable for publication in PLOS ONE. Congratulations! Your manuscript is now with our production department. 

Kind regards, 

on behalf of

Professor Sascha Köpke 

Academic Editor

PLOS ONE